# Thermodynamic Guidelines for the Mechanosynthesis or Solid-State Synthesis of MnFe_2_O_4_ at Relatively Low Temperatures

**DOI:** 10.3390/ma17020299

**Published:** 2024-01-07

**Authors:** Isabel Antunes, Miguel F. Baptista, Andrei V. Kovalevsky, Aleksey A. Yaremchenko, Jorge R. Frade

**Affiliations:** Department of Materials and Ceramics Engineering, CICECO-Aveiro Institute of Materials, University of Aveiro, 3810-193 Aveiro, Portugal; m.f.b@ua.pt (M.F.B.); akavaleuski@ua.pt (A.V.K.); ayaremchenko@ua.pt (A.A.Y.)

**Keywords:** manganese ferrite, mechanosynthesis, solid-state synthesis, thermodynamic guidelines

## Abstract

Herein, thermodynamic assessment is proposed to screen suitable precursors for the solid-state synthesis of manganese ferrite, by mechanosynthesis at room temperature or by subsequent calcination at relatively low temperatures, and the main findings are validated by experimental results for the representative precursor mixtures MnO + FeO_3_, MnO_2_ + Fe_2_O_3_, and MnO_2_ +2FeCO_3_. Thermodynamic guidelines are provided for the synthesis of manganese ferrite from (i) oxide and/or metallic precursors; (ii) carbonate + carbonate or carbonate + oxide powder mixtures; (iii) other precursors. It is also shown that synthesis from metallic precursors (Mn + 2Fe) requires a controlled oxygen supply in limited redox conditions, which is hardly achieved by reducing gases H_2_/H_2_O or CO/CO_2_. Oxide mixtures with an overall oxygen balance, such as MnO + Fe_2_O_3_, act as self-redox buffers and offer prospects for mechanosynthesis for a sufficient time (>9 h) at room temperature. On the contrary, the fully oxidised oxide mixture MnO_2_ + Fe_2_O_3_ requires partial reduction, which prevents synthesis at room temperature and requires subsequent calcination at temperatures above 1100 °C in air or in nominally inert atmospheres above 750 °C. Oxide + carbonate mixtures, such as MnO_2_ +2FeCO_3_, also yield suitable oxygen balance by the decomposition of the carbonate precursor and offer prospects for mechanosynthesis at room temperature, and residual fractions of reactants could be converted by firing at relatively low temperatures (≥650 °C).

## 1. Introduction

The Mn−Fe−O system is based on abundant transition metal elements and offers rich redox flexibility in mixed-oxide compounds, such as spinel compositions $({Mn,Fe)}_{3}O_{4}$ or bixbyite $({Mn,Fe)}_{2}O_{3}$. Low-cost precursors combined with affordable processing methods may broaden the applicability of Mn−Fe−O materials from up-to-date applications of ferrites in nanotechnologies to mass production processes such as the chemical looping gasification of biomass [1], oxygen storage materials for chemical looping combustion [2], pigments [3], etc., while also extending the applicability of corresponding single-oxide Fe−O and Mn−O systems [4]. Available low-grade Fe- and Mn-rich ores also contribute to the feasibility of some of these applications of manganese ferrite [3]. In this case, one should also consider the role of gangue components of the natural precursor, which might affect relevant properties and/or may segregate as secondary crystalline phases or in the amorphous fraction after high-temperature firing.

Phase equilibria in the Mn−Fe−O system depend on the Mn:Fe ratio and thermochemical conditions of processing, i.e., redox conditions and temperature, as described by phase diagrams [5] and thermodynamic modelling [6]. Thus, prospects to obtain single-phase $Mn{Fe}_{2}O_{4}$ depend on firing schedules and specific atmospheres such as ${CO}_{2}/CO$ [7], which may induce unusual microstructures and potential impacts on the magnetic and dielectric properties of ferrites. The effective phases obtained by solid-state reactions may also depend on starting precursors [8]. 

Conditions of processing, combined with subsequent heat treatments [9,10,11], also determine the crystallite size, changes in oxidation states, and cation occupancy in tetrahedral and octahedral positions of ${Mn}_{x}{Fe}_{3-x}O_{4}$ spinels, with corresponding effects on relevant properties and prospective applications [12]. In addition, thermochemical treatments may cause the selective segregation of iron oxides or manganese oxides [13] and corresponding changes in the Mn:Fe ratio of nonstoichiometric spinels ${Mn}_{x}{Fe}_{3-x}O_{4}$. 

Thus, it is important to develop flexible methods allowing the synthesis of ferrites in wide ranges of mechanochemical and/or thermochemical conditions, including processing at room temperature by direct mechanosynthesis from solid precursor mixtures, to avoid undue changes promoted at higher temperatures, depending on suitable precursors [14]. Mechanochemical treatments may also induce nanostructuring and changes in site occupancy [15], including the nanostructuring of $Mn{Fe}_{2}O_{4}$ after previous firing at high temperature [16].

The high density and hardness of steel milling vials and balls are most suitable for high-energy milling to overcome the kinetic limitations of room-temperature solid-state kinetics. However, these milling media may induce contamination with metallic Fe, changing the intended Fe:Mn stoichiometry, or causing the onset of secondary mixed-oxide phases, such as (Fe,Mn)O [17]. The onset of metallic Fe was also reported after the mechanosynthesis of a ${0.5Mn}_{2}O_{3}+{Fe}_{2}O_{3}$ powder mixture, in spite of its stoichiometric ratio (Mn:Fe=1:2) and oxygen excess relative to the spinel structure, i.e., $O:\left(Mn+Fe\right)>4:3$ [18]. Differently, $MnO+{Fe}_{2}O_{3}$ or ${1/3Mn}_{3}O_{4}+{Fe}_{2}O_{3}$ powder mixtures did not react under mild mechanical activation, and conversion to manganese ferrite required calcination at about 900 °C in an inert atmosphere [19]. Metallic Fe+Mn precursors were used as precursors for the mechanosynthesis of $Mn{Fe}_{2}O_{4}$ by wet milling [20], and metal + oxide precursor mixtures were also proposed, such as ${2Fe+MnO}_{2}$ [9] or ${Mn+Mn}_{2}O_{3}+3{Fe}_{2}O_{3}$ [21]. On the contrary, the mechanical activation of fully oxidised precursors ($MnO_{2}$ + ${Fe}_{2}O_{3}$) only induced amorphisation, even after 40 h, and conversion to $Mn{Fe}_{2}O_{4}$ required subsequent calcination at high temperatures in the order of 1200 °C [22], except possibly in relatively reducing conditions [7] or a vacuum [21].

Non-oxide precursors have also been proposed such as mixtures of $Mn{\left(OH\right)}_{2}+{Fe}_{2}O_{3}$ or $Mn{\left(OH\right)}_{2}$+$Fe{\left(OH\right)}_{3}$. Both mixtures converted to $Mn{Fe}_{2}O_{4}$ after mechanosynthesis in a steel vial and with steel balls for 25 h [23]. The onset of intermediate phases after a shorter milling time shows that the stable oxyhydroxide phase FeOOH forms readily under mechanical activation by the decomposition of $Fe{\left(OH\right)}_{3}$ or partial hydration of ${Fe}_{2}O_{3}$ in contact with $Mn{\left(OH\right)}_{2}$.

Mixed carbonate + oxide precursor mixtures were also proposed, namely $Mn{CO}_{3}+{Fe}_{2}O_{3}$, which were milled at 450 rpm in toluene, for as long as 90 h, and required subsequent calcining at 750 °C [24] or other combinations of mechanical activation and calcination [25], for conversion to $Mn{Fe}_{2}O_{4}$. The $2Fe{CO}_{3}+MnO_{2}$ powder mixture was found more successful in reaching conversion to the spinel phase at room temperature, within the detection limits of X-ray diffraction, by direct mechanosynthesis at a sufficiently high milling rate, even without subsequent calcination [26].

Other precursors were also proposed for the mechanochemical synthesis of $Mn{Fe}_{2}O_{4}$, such as the milling of nitrate mixtures with citric acid and subsequent calcination at 300−400 °C [27], or the mechanically assisted disproportionation reaction of $2Fe{Cl}_{3}+4MnO\to Mn{Fe}_{2}O_{4}+3Mn{Cl}_{2}$ [28].

Thus, the main objective of this work was to propose guidelines for the room-temperature mechanosynthesis of $Mn{Fe}_{2}O_{4}$ from different combinations of precursor mixtures or by mechanical activation and subsequent calcination at intermediate temperatures, avoiding tedious trial and error tests and minimizing undue experimental work. This was guided by thermodynamic predictions and demonstrated by experimental evidence using representative mixtures of precursors.

## 2. Methods

### 2.1. Thermodynamic Modelling

Quasi-planar chemical potential diagrams for mixed metal-oxide systems were derived analogously to the method that was proposed by Yokokawa and co-authors [29,30], previously applied to assess the phase stability in a variety of systems in solid-state electrochemical applications [31], and to assess their contamination [30,32] or degradation by interaction with reactive gases [33]. Other recent references address relevant issues of materials proposed for catalytic applications in different systems such as Ni−Al−O [34], Fe−Ti−O [35], $CaO-SiO-CO_{2}$ [36], etc., or the synthesis of catalytic materials in the systems Fe−Si−O [37], Ca−Fe−O [38], or Ce−Al−O [39], and guidelines for the processing of $SrTiO_{3}$ by mechanosynthesis [40].

Similar methods were applied here as guidelines for the mechanosynthesis of $Mn{Fe}_{2}O_{4}$ from different combinations of precursors in the systems Mn−Fe−O  or Mn−Fe−O−C. Chemical potential diagrams were derived from 2-phase equilibria, which establish relations between the chemical potentials of the transition elements, the chemical potentials of $O_{2}$ and/or $CO_{2}$, and the free energy of the corresponding reactions. One may refer to the $MnCO_{3}/Mn{Fe}_{2}O_{4}$ equilibrium as a representative example:(1)$$Mn+0.5{MnFe}_{2}O_{4}+1.5{CO}_{2}⟺Fe+1.5MnCO_{3}+1/4O_{2};$$
(2)$$∆\mu_{Mn}-∆\mu_{Fe}=RTln\left(a_{Mn}/a_{Fe}\right)=∆G+1/4RTln\left({pO}_{2}\right)-1.5RTln\left({pCO}_{2}\right).$$

These are expressed per unit of Mn and unit of Fe to obtain a ready relation between their activity ratio and the free energy of the reaction. In this case, we obtained planar diagrams for the dependence of the chemical potentials of transition elements $RTln\left(a_{Mn}/a_{Fe}\right)$ vs. the chemical potential of oxygen $RTln\left({pO}_{2}\right)$, for a specified partial pressure of $CO_{2}$. Thermodynamic data required for the calculations of free energy were retrieved from the database of the FACTSAGE v.5.5 software package [41].

These chemical potential diagrams can be a suitable guideline for the solid-state kinetics of the formation of generic spinels $AB_{2}O_{4}$, which has been related to the chemical potential gradient of the controlling species across the reaction product, namely $∆\mu_{A}$, for conditions when the controlling species is $A^{n+}$ (Figure 1a,b) or $-∆\mu_{B}$ for $B^{m+}$ **(**Figure 1c,d) [42]. In fact, the controlling cationic species may also depend on the redox conditions and tetrahedral or octahedral site occupancy, with variable degrees of inversion, and possibly also deviations from the nominal trivalent/divalent ratio, $\left\{\left[{Fe}^{3+}\right]+\left[{Mn}^{3+}\right]\right\}:\left\{\left[{Fe}^{2+}\right]+\left[{Mn}^{2+}\right]\right\}>2:1$ [43]; this may be compensated by cation vacancies $({Fe,Mn)}_{3-\delta }O_{4}$ [44]. High-energy milling or subsequent firing may also induce variable degrees of vacancy ordering [45], with an impact on diffusivity. 

Thus, one must consider the combined differences $∆\mu_{A}-∆\mu_{B}$ to account for both cases of controlling cationic species or their combination, assuming that charge neutrality is readily sustained by electronic conductivity, relying on the hopping of polarons [46]. In addition, porous samples are expected to allow ready oxygen transport, minimising local gradients of the chemical potential of oxygen ($∆\mu_{O_{2}}\approx 0)$. Otherwise, one should consider the co-diffusion of cationic and anionic species, which also depends on $∆\mu_{O_{2}}$ (Figure 1b,d). In fact, the diffusivities of different cationic species in spinels [47] are expected to exceed the diffusivity of oxygen ions [48], at least at high temperatures (≥1200 °C). 

### 2.2. Experimental Methods

The $MnO+{Fe}_{2}O_{3}$ (Sigma Aldrich, St. Louis, MO, USA, 99%) powder mixture was used as a case study to demonstrate the mechanosynthesis of $Mn{Fe}_{2}O_{4}$ at room temperature from precursor mixtures with oxygen balance. MnO was obtained by the reduction of $MnO_{2}$ (Alfa Aesar, Ward Hill, MA, USA, 99%) in flowing 10% $H_{2}$–90% N_2_ at 850 °C for 5 h. Mechanosynthesis was performed in air, in a PM100 Retsch planetary ball mill, using a TZP vial (Retsch, Haan, Germany, 125 cm^3^) and TZP balls (TOSOH Co., Tokyo, Japan) of diameters 10 mm and 15 mm at a ratio of 2:1, with a balls/powder weight ratio of 14:1, at a rotational speed of 500 rpm, and for 3 or 9 h of cumulative milling time. Milling was performed for periods of 10 min with a subsequent pause of 5 min to prevent overheating by attrition. Powder X-ray diffraction (XRD) was performed for the phase identification of precursors and reacted mixtures after mechanosynthesis, using a Rigaku SmartLab SE diffractometer (Rigaku, Tokyo, Japan , in the X-ray fluorescence reduction mode of a 2D detector with a CuK_α_ radiation source (40 kV, 30 mA), in the 2θ range 10−90°, with a step size = 0.02°, and with a speed of 3 °/min. Nickel powder (Alfa Aesar, 99.9%) was added as an internal reference to quantify phase contents.

The fully oxidised oxide mixture $MnO_{2}+{Fe}_{2}O_{3}$ was selected as a case study of precursors which require a combination of mechanical activation and subsequent thermochemical treatment. In this case, mechanical activation was performed in a TZP vial, with TZP zirconia balls of diameter 10 mm and with a balls/powder weight ratio of 10:1, at 350 or 450 rpm, for 10 h of cumulative milling time, with periods of 5 min milling followed by a 5 min pause. Subsequent thermal treatments of activated powders were performed in air and in argon atmospheres in the temperature range of 750 to 1000 °C. XRD was used for phase identification in as-milled precursors and after thermal treatments.

A mixture of ${MnO}_{2}+FeCO_{3}$-based natural siderite was used as a case study of mechanosynthesis from an oxide + carbonate precursor mixture, under different milling conditions, as shown in Table 1. Several samples which were mechanically activated at 450 rpm, were subsequently treated at temperatures from 550 °C to 900 °C for times ranging from 2 h to 8 h in an Ar atmosphere. The cation composition of the natural siderite was assessed by inductively coupled plasma optical emission spectroscopy (ICP-OES) (Jobin Yvon Activa M, Horiba, Kyoto, Japan), yielding: 70.5% Fe, 18.6% Mg, 5.6% Ca, 3.4% Si, and 1.9% Al.

## 3. Thermodynamic Guidelines

### 3.1. Synthesis from Oxide and/or Metallic Precursors

Figure 2 shows the chemical potential diagram for metal-oxide phases in the Mn−Fe−O system at room temperature, including the driving forces for the solid-state reaction from the stoichiometric $MnO+{Fe}_{2}O_{3}$ mixture; this confirms the prospects of room-temperature mechanosynthesis. The driving force is expected to converge to $∆\mu_{Mn}-∆\mu_{Fe}\approx \;$35 kJ/mol, which corresponds to the chemical potential differences between the three-phase contacts ${MnO/Mn}_{3}O_{4}/Mn{Fe}_{2}O_{4}$ and ${Fe}_{3}O_{4}/{Fe}_{2}O_{3}/{MnFe}_{2}O_{4}$, as shown in Figure 2, for sufficiently porous samples with the ready transport of $O_{2}$ and minimum gradients of the chemical potential of oxygen $∆\mu_{O_{2}}.$

In addition, the ${MnO+Fe}_{2}O_{3}$ powder mixture yields a precise oxygen balance in contact with $O_{2}$-lean atmospheres:(3)$$MnO+{Fe}_{2}O_{3}\to Mn{Fe}_{2}O_{4}.$$
The free energy $∆G_{R}=-25\;kJ/mol$ shows that this reaction is spontaneous, assuming self-controlled redox conditions, as expected in sealed or inert atmospheres or under moderate vacuum [17]. In fact, the redox pairs ${MnO/Mn}_{3}O_{4}$ and ${{Fe}_{3}O_{4}/Fe}_{2}O_{3}$ provide nearly stable redox conditions, at nearly identical chemical potentials of $O_{2}$, and may act as temporary redox buffers in nominally inert atmospheres (e.g., Ar) or even in air. In these conditions, one may assume that the $MnO/Mn{Fe}_{2}O_{4}$ equilibrium converges towards the $MnO-Mn{Fe}_{2}O_{4}-{Mn}_{3}O_{4}$ triple point, and ${Fe}_{2}O_{3}/Mn{Fe}_{2}O_{4}$ equilibrium converges towards the ${Fe}_{2}O_{3}-Mn{Fe}_{2}O_{4}-{Fe}_{3}O_{4}$ triple point. 

Oxygen balance is also expected for the synthesis of $Mn{Fe}_{2}O_{4}$ from mixtures of ${Mn}_{3}O_{4}$ and ${Fe}_{3}O_{4}$:(4)$$1/3{Mn}_{3}O_{4}+2/3{Fe}_{3}O_{4}\to Mn{Fe}_{2}O_{4};$$
with the free energy of the reaction $∆G_{R}=-24\;kJ/mol$. In fact, the ${1/3Mn}_{3}O_{4}$+${2/3Fe}_{3}O_{4}$ precursor mixture may even provide better kinetic conditions for the mechanosynthesis of $Mn{Fe}_{2}O_{4}$ as compared to the $MnO+{Fe}_{2}O_{3}$ mixture, based on the structural similarity of ${Fe}_{3}O_{4}$ and $Mn{Fe}_{2}O_{4}$ (space group $Fd\bar{3}m$). Thus, one expects gradual concentration profiles across a diffuse interface ${Fe}_{3}O_{4}$/${MnFe}_{2}O_{4}$ boundary, which is represented by a dotted line in Figure 2. In addition, the Mn:Fe ratio in manganese ferrite may deviate from the nominal stoichiometry, i.e., ${Mn}_{1+x}{Fe}_{2+x}O_{4}$. A clear boundary with a sharp concentration change is only expected at the ${MnFe}_{2}O_{4}$/${Mn}_{3}O_{4}$ interface, taking into account the structural differences between the cubic structure of $Mn{Fe}_{2}O_{4}$ (space group $Fd\bar{3}m$) and the tetragonal structure of ${Mn}_{3}O_{4}$ (space group $I4_{1}/amd$). 

The $MnO_{2}+{2Fe}_{2}O$ powder mixture also allows oxygen balance:(5)$$MnO_{2}+2FeO\to Mn{Fe}_{2}O_{4}$$
and the free energy of this reaction $∆G_{R}=-167\;kJ/mol$ indicates that this should be even more spontaneous than Equations (3) or (4). However, the mechanisms of this reaction may be complex since FeO is a very unstable phase and may oxidise readily to ${Fe}_{3}O_{4}$ in contact with the strongly oxidising phase $MnO_{2}$, which may also undergo gradual reduction through intermediate phases (${Mn}_{2}O_{3}$ and ${Mn}_{3}O_{4}$) and, eventually, down to the ${Mn}_{3}O_{4}$/${MnFe}_{2}O_{4}$ interface in Figure 2. In fact, the two-phase ${Fe}_{3}O_{4}$/${Fe}_{2}O_{3}$ redox pair is strongly reducing relative to the redox pairs ${MnO}_{2}$/${Mn}_{2}O_{3}$ and ${Mn}_{3}O_{4}$/${Mn}_{2}O_{3}$ and less reducing relative to the redox conditions of the three-phase contact ${Mn}_{3}O_{4}$/${Fe}_{2}O_{3}/Mn{Fe}_{2}O_{4}$.

Other oxide mixtures require different approaches to meet the oxygen balance. Ding et al. [21] reported the synthesis of a $Mn{Fe}_{2}O_{4}$-based spinel by the mechanical activation of ${Mn}_{2}O_{3}+{Fe}_{2}O_{3}$ powder mixtures in Ar atmosphere:(6)$$0.5{Mn}_{2}O_{3}+{Fe}_{2}O_{3}\to Mn{Fe}_{2}O_{4}+0.25O_{2}.$$

In this case, the free energy of this reaction $∆G_{R}=+67\;kJ/mol$ shows that the inert atmosphere (Ar) does not meet this requirement at room or intermediate temperature (Figure 3a) since the redox conditions of Ar only reach the stability window of $Mn{Fe}_{2}O_{4}$ at temperatures above about 800 °C (Figure 3b). Firing in air would require even higher temperatures, namely above 1100 °C (Figure 3c); this suggests an apparent contradiction between the actual thermodynamic predictions and the reported experimental results obtained in Ar or air at a relatively low temperature [21]. In this case, one may assume that the oxygen balance was reached by the incorporation of a fraction of metallic Fe from the milling balls and vial, as mentioned by others [18], i.e.:(7)$$1/3Fe+4/9{Mn}_{2}O_{3}+8/9{Fe}_{2}O_{3}\to {Mn}_{8/9}{Fe}_{19/9}O_{4}.$$

A similar compensation to reach the oxygen balance may be provided by a fraction of Mn, as reported also by Ding et al. [21]:(8)$$1/3Mn+{1/3Mn}_{2}O_{3}+{Fe}_{2}O_{3}\to Mn{Fe}_{2}O_{4}.$$
The free energy of this reaction $∆G_{R}=-95\;kJ/mol$ also shows that this should be spontaneous, except for difficulties raised by the high instability of the metallic precursor.

Figure 3 also shows that even higher firing temperatures should be required for the synthesis of manganese ferrite from the most stable oxide mixture (${MnO}_{2}+{Fe}_{2}O_{3}$):(9)$${MnO}_{2}+{Fe}_{2}O_{3}\to Mn{Fe}_{2}O_{4}+0.5O_{2}.$$

In this case, the free energy $∆G_{R}=+108\;kJ/mol$ shows that reactivity should require highly reducing conditions or the synthesis of $Mn{Fe}_{2}O_{4}$ in air at high temperatures, i.e., 1100 °C [49] or higher temperatures [8]. Inert atmospheres or a vacuum may allow synthesis at somewhat lower temperatures but still in the order of 800 °C.

Figure 3 also emphasises the limited temperature range when the redox pair $H_{2}/H_{2}O$ may be used to adjust the thermochemical conditions of $Mn{Fe}_{2}O_{4}$. This shows that $H_{2}$-based gases only provide feasible oxygen balance for synthesis from ${Mn}_{2}O_{3}+{Fe}_{2}O_{3}$ or ${MnO}_{2}+{Fe}_{2}O_{3}$ from room temperature to about 773 K, depending on the $H_{2}:H_{2}O$ ratio, and from $H_{2}$-rich conditions (e.g., $H_{2}:H_{2}O=100$) to $H_{2}$-lean conditions ($H_{2}:H_{2}O=0.01$). Thus, the $H_{2}/H_{2}O$ pair also fails as a redox buffer at intermediate temperatures in the range of 500 °C−800 °C. Still, the $H_{2}:H_{2}O$ redox buffer may explain the feasibility of the direct mechanosynthesis of $Mn{Fe}_{2}O_{4}$ from metallic precursors in wet conditions [20], as follows:(10)$$Mn+2Fe+4H_{2}O⟺Mn{Fe}_{2}O_{4}+4H_{2}.$$
Note that this reaction should be highly spontaneous in a wet hydrogen atmosphere ($∆G_{R}=-204\;kJ/mol$). A fraction of metallic precursor added to mixtures of $MnO_{2}$ and ${Fe}_{2}O_{3}$ may also allow oxygen balance by the partial substitution of hematite:(11)$${MnO}_{2}+2/3Fe+2/3{Fe}_{2}O_{3}⟺Mn{Fe}_{2}O_{4};$$
($∆G_{R}=-171\;kJ/mol$) or by changing the Mn:Fe ratio in the manganese ferrite:(12)$$0.6Fe+0.8{MnO}_{2}+0.8{Fe}_{2}O_{3}⟺{Mn}_{4/5}{Fe}_{11/5}O_{4}.$$

This metallic fraction may be introduced by the erosion of steel milling media [17,18].

### 3.2. Synthesis from Carbonate + Carbonate or Carbonate + Oxide Precursor Mixtures

Figure 4 shows the stability ranges of iron carbonate and manganese carbonate and their equilibrium with the oxide phase or metallic Fe. This emphasises that the high-energy milling of carbonate mixtures is unlikely to yield manganese ferrite at room temperature due to the wide stability range of manganese carbonate, even when the partial pressure of $CO_{2}$ remains low (Figure 4a). An onset of manganese ferrite from carbonate precursors and under a $CO_{2}$-rich atmosphere is only expected upon heating to temperatures above 560 K, as also shown in Figure 4c, or somewhat lower temperatures in a $CO_{2}$-lean atmosphere, such as above 473 K with $p{CO}_{2}\approx 0.01$ atm, after previous decomposition of iron carbonate. In fact, the redox range of $FeCO_{3}$ stability in Figure 4 narrows rapidly upon approaching this temperature (Figure 4b), in close agreement with experimental evidence that the decomposition of $FeCO_{3}$ occurs likely upon heating above about 200 °C for synthetic $FeCO_{3}$ or at higher temperatures for low-grade natural siderite minerals [50]. In this case, the decomposition of siderite only retains the divalent state of iron as a thermodynamically unstable phase at intermediate temperatures, evolving to ${Fe}_{3}O_{4}+Fe$ in an inert atmosphere, and progressing by extinction of the fraction of metallic Fe and subsequent oxidation to magnetite or hematite in sufficiently oxidising atmospheres.
(13)$$4Fe{CO}_{3}\to 4FeO+{4CO}_{2}↑\to Fe+{Fe}_{3}O_{4}\overset{O_{2}}{\to }1.5{Fe}_{3}O_{4}\overset{O_{2}}{\to }2{Fe}_{2}O_{3}.$$

The stability ranges of manganese carbonate [51] and iron carbonate [52] and their decomposition temperatures also depend on operating atmospheres and may be delayed by kinetic limitations. For example, the decomposition of $MnCO_{3}$ in atmospheres with a moderate partial pressure of $CO_{2}$ (≈20 kPa) and upon heating at 6 K/min reached a peak at ≈ 685 K, and the corresponding activation energy was about 270 kJ/mol [51]. From these parameters, one may predict a decomposition peak at ≈ 607 K upon heating at a very low rate (0.1 K/min); this is close to thermodynamic predictions for the decomposition of manganese carbonate ($Mn{CO}_{3}\to MnO+{CO}_{2}$) at T>607 K under ${pCO}_{2}\approx 0.2\;atm$. Thus, one may expect the conversion of both carbonate precursors on heating to intermediate temperatures, and the stability diagram should converge to the corresponding oxide phases (Figure 3).

Figure 4 also shows that the synthesis of manganese ferrite by the mechanical activation of $Mn{CO}_{3}+{Fe}_{2}O_{3}$ powder mixtures should require subsequent heating to intermediate temperatures at the onset of a stability window for $Mn{Fe}_{2}O_{4}$. In fact, synthesis has been reported after the mechanical activation of $Mn{CO}_{3}+{Fe}_{2}O_{3}$ precursors and then calcining at 673 K [25], which is somewhat higher than the decomposition temperature predicted for $Mn{CO}_{3}\to MnO+{CO}_{2}$ in a $CO_{2}$ atmosphere (≈ 649 K). Thus, one cannot exclude the possibility of the onset of MnO formed as an intermediate phase before the conversion of the resulting oxide mixture to $Mn{Fe}_{2}O_{4}$, as described by Equation (3).

Room-temperature mechanosynthesis of $Mn{Fe}_{2}O_{4}$ from manganese dioxide and iron carbonate was demonstrated recently [26]; this may seem to contradict the thermodynamic equilibrium predictions at room temperature in a $CO_{2}$-rich atmosphere (Figure 5a), which suggests thermodynamic feasibility for the onset of $MnCO_{3}$ by the partial reduction of $MnO_{2}$, combined with the oxidative decomposition of $FeCO_{3}$:(14)$$Fe{CO}_{3}+{MnO}_{2}\to 1/2{Fe}_{2}O_{3}+Mn{CO}_{3}+1/4O_{2}$$
with $∆G_{R}=-58\;kJ/mol$. However, the carbonation of manganese oxides may be hindered by kinetic limitations, except for the ready carbonation of MnO [53], requiring sufficiently reducing conditions to reach the redox range of MnO, i.e., $RTln\left(pO_{2}\right)<-387\;kJ/mol$ at room temperature. In the reported conditions of mechanical activation of the ${FeCO}_{3}+MnO_{2}$ reactants [26], the reducing conditions should be established by the $Fe{CO}_{3}{/Fe}_{2}O_{3}$ redox pair, reaching only $RTln\left(pO_{2}\right)\approx -322\;kJ/mol$ in a $CO_{2}$ atmosphere (Figure 5). Thus, we devised an alternative metastable diagram (dashed lines in Figure 5), which is based on metastable two-phase boundaries in the absence of manganese carbonate; this includes the $Fe{CO}_{3}{/Fe}_{2}O_{3}$ boundary, combined with metastable two-phase boundaries $Fe{CO}_{3}{/Mn}_{3}O_{4}$ and $Fe{CO}_{3}{/MnFe}_{2}O_{4}$, and the oxide/oxide boundaries expected for the Mn−Fe−O system in the moderate-to-oxidising range. This metastable diagram provides a plausible explanation for the onset of $Mn{Fe}_{2}O_{4}$ at room temperature, assuming that the conditions of high-energy milling are sufficient to overcome kinetic restrictions. Note also that the metastable window of $Mn{Fe}_{2}O_{4}$ may even be enlarged by slight overheating (Figure 5b), which may be caused by attrition during mechanical activation.

### 3.3. Synthesis from Hydroxide + Oxide or Hydroxide + Oxyhydroxide Precursor Mixtures

The mechanosynthesis of manganese ferrite from hydroxide $Mn{\left(OH\right)}_{2}+2Fe{\left(OH\right)}_{3}$ mixtures or hydroxide + oxide $Mn{\left(OH\right)}_{2}+{Fe}_{2}O_{3}$ mixtures was also clearly demonstrated [23]. However, the chemical potential diagrams (Figure 6a) show that the stable phases in contact with manganese ferrite should be $Mn{\left(OH\right)}_{2}$ and FeOOH, because the $Fe{\left(OH\right)}_{3}$ phase is unstable and transforms readily to goethite FeOOH during the high-energy milling of hydroxide mixtures:(15)$$Fe{\left(OH\right)}_{3}\to FeOOH+H_{2}O.$$

The stability of the goethite phase may also explain its onset as an intermediate phase, combined with MnO [23], by the mechanical activation of mixtures of manganese hydroxide + hematite:(16)$$Mn{\left(OH\right)}_{2}+{Fe}_{2}O_{3}\to MnO+2FeOOH.$$

Nevertheless, the stability range of $Mn{Fe}_{2}O_{4}$ in equilibrium with the hydroxide + oxyhydroxide mixture was still very narrow at room temperature (Figure 6a), even in fairly dry conditions and only up to about $pH_{2}O\approx 0.03\;atm$, when the chemical potential driving force $∆\mu_{Mn}-∆\mu_{Fe}$ vanished. Thus, one may be surprised by the evidence that $Mn{Fe}_{2}O_{4}$ still formed after a sufficiently long time (25 h) in air. A plausible explanation for this experimental evidence may be based on slight overheating under high-energy milling, taking into account that the stability range of $Mn{Fe}_{2}O_{4}$ increases significantly with slight heating (Figure 6b). Note that the authors of ref. [23] did not mention if milling was stopped at regular intervals to prevent overheating by attrition. 

## 4. Experimental Validation

### 4.1. Synthesis from Oxide Mixtures

Figure 7 shows the feasibility of the mechanosynthesis of $Mn{Fe}_{2}O_{4}$ (JCPDS 01-074-2403) from stoichiometric powder mixtures of MnO (JCPDS 01-075-1090) +${Fe}_{2}O_{3}$ (JCPDS 01-087-1166) at a moderate milling rate of 500 rpm for 3 h to 9 h. Though large fractions of reactants are still present after milling for a relatively short milling time, one already observes significant intensity in the main reflections of the spinel phase after 3 h, using Ni (JCPDS 01-087-0712) as reference. The intensities of precursor phases then drop markedly after milling for 9 h, and the spinel phase becomes prevailing, co-existing with residual contents of reactants (${Fe}_{2}O_{3}$ and MnO) and the onset of magnetite  ${Fe}_{3}O_{4}$e (JCPDS 01-079-0418). This confirms the three-phase contact $Mn{Fe}_{2}O_{4}/{Fe}_{2}O_{3}/{Fe}_{3}O_{4}$, as indicated in Figure 2. On the contrary, one did not detect ${Mn}_{3}O_{4}$ (JCPDS 01-080-0382) and could not confirm the three-phase contact $Mn{Fe}_{2}O_{4}/MnO/{Mn}_{3}O_{4}$ in Figure 2. Still, one may assume that the redox condition remains reducing and is nearly controlled by the ${Fe}_{2}O_{3}/{Fe}_{3}O_{4}$ redox pair; this retains conditions for reactivity at room temperature, and the kinetics are mainly determined by the chemical potential gradients of the transition metal elements ($∆\mu_{Mn}{-∆\mu }_{Fe}$) shown in Figure 2.

The gradual conversion of precursors was also assessed by the reference intensity ratio (RIR) [54] based on the integrated intensities (Figure 7) of the main reflections of ${Fe}_{2}O_{3}$ ($I_{{Fe}_{2}O_{3}\left(104\right)}$) and MnO ($I_{MnO\left(200\right)}$), taking the main reflection of Ni ($I_{Ni\left(111\right)}$) as a reference. The Ni standard was added as 20 *wt*.% of the reactants’ contents in the stoichiometric mixture of $MnO+{Fe}_{2}O_{3}.$ The composition of this three-phase mixture $(26.6\;wt.\%MnO+57.7\;wt.\%{Fe}_{2}O_{3}+16.7\;wt.\%Ni$) was then combined with their intensity ratios to obtain the calibration factors, and these were used to assess the contents of residual reactants after 3 h and after 9 h, as shown in Table 2.

The fully oxidised mixture ${MnO}_{2}$ (JCPDS 00-024-0735) +${Fe}_{2}O_{3}$ is not reactive at room temperature (Figure 8). In fact, Figure 2 shows a large gap between the redox range of the reacting ${MnO}_{2}$+${Fe}_{2}O_{3}$ mixture and the stability range of $Mn{Fe}_{2}O_{4}$ ($∆\mu_{O_{2}}\geq 134\;kJ/mol$). Mechanical activation was performed at 350 rpm or 450 rpm for 10 h, and changes in milling conditions only yielded a decrease in the intensity of reactant reflections. The synthesis of $Mn{Fe}_{2}O_{4}$ required subsequent thermochemical treatment at sufficiently high temperatures to overcome the redox gap, such as heating to about 1000 °C in air, as indicated by the thermodynamic predictions (Figure 2), and firing at lower temperatures converted the $MnO_{2}$ precursor to a bixbyite phase, which may contain significant contents of Fe, i.e., $({Mn,Fe)}_{2}O_{3}$, in close agreement with thermodynamic modelling [5]. Otherwise, Figure 3 suggests that firing in an inert (Ar) atmosphere may allow the onset of $Mn{Fe}_{2}O_{4}$ at ≈800 °C. In fact, deviations from stoichiometry may explain the results obtained by firing at 750 °C in Ar, which already showed the onset of a spinel phase, co-existing with ${Fe}_{2}O_{3}$ and traces of ${Mn}_{2}O_{3}$ (JCPDS 01-076-0150). The corresponding thermodynamic predictions suggest that $Mn{Fe}_{2}O_{4}$ should co-exist with ${Fe}_{2}O_{3}$ and ${Mn}_{3}O_{4}$ rather than ${Mn}_{2}O_{3}$. A fraction of residual ${Fe}_{2}O_{3}$ was still present after firing at 900 °C (Figure 8), indicating kinetic limitations and suggesting that the composition of the spinel phase still deviates from equilibrium, i.e., ${Mn}_{1+\delta }{Fe}_{2-\delta }O_{4}$.

### 4.2. Synthesis from Carbonate + Oxide Mixtures

Figure 9 shows the results of the mechanical activation or mechanosynthesis of mixtures of $MnO_{2}+FeCO_{3}$-based siderite achieved under the different conditions detailed in Table 1. We found significant effects of the milling rate, which caused a decrease in the reflections of both precursors and showed an onset of the main reflections of the spinel phase upon changing from 350 rpm to 450 rpm. The milling media also played a key role in the structural changes since the use of mild nylon vial hinders conversion to the spinel phase and may even enhance the crystallinity of the carbonate phase. On the contrary, the use of the TZP vial and more severe milling conditions allowed a complete extinction of the XRD reflections of both precursors. In fact, the free energy $∆G_{R}=-80\;kJ/mol$ suggests prospects for complete conversion to the spinel:(17)$$Fe{CO}_{3}+{MnO}_{2}\to 1/2{Fe}_{2}O_{3}+Mn{CO}_{3}+1/4O_{2}.$$

Thus, mechanosynthesis can be attained by sufficiently high-energy milling, achieved with a high balls/powder ratio, a fraction of heavier balls, or a combination of higher rotation speed and longer time [26].

Still, the reflections of the spinel phase are located between the corresponding reflections of $Mn{Fe}_{2}O_{4}$ (dashed lines) and ${Fe}_{3}O_{4}$ (dotted lines), suggesting partial Mn-deficiency, i.e., ${Mn}_{1-x}{Fe}_{2+x}O_{4}$. Note that magnetite and manganese ferrite possess identical structures. In addition, a fraction of the initial precursors may be incorporated in a significant amorphous phase [26], possibly combined with the gangue components from the natural siderite precursor, which contains relatively large fractions of Mg and Ca. These impurities may exert significant effects on the structural features and properties of manganese ferrite [55,56].

Subsequent heat treatments at intermediate temperatures were planned to confirm that the conversion of $MnO_{2}+FeCO_{3}$-based siderite to $Mn{Fe}_{2}O_{4}$ is hindered mainly by kinetics. The prevailing conversion to the spinel only occurred at temperatures ≥ 900 °C, whereas intermediate temperatures yielded mainly intermediate phases:.
(18)$${MnO}_{2}+2Fe{CO}_{3}\to 1/3{Mn}_{3}O_{4}+2/3{Fe}_{3}O_{4}+2{CO}_{2}$$

In fact, we expected a ready decomposition of both precursors upon heating [50,51] before reaching the lowest firing temperature in Figure 10 (550 °C). Heat treatments at higher temperatures may be controlled by a gradual conversion of the intermediate oxide phase to the spinel phase, as emphasised by the results obtained by firing at 650 °C for different times (Equation (4)).

In this case, one may assume that the driving force for the synthesis of $Mn{Fe}_{2}O_{4}$ corresponds to the chemical potential differences between the two-phase lines ${Fe}_{3}O_{4}/Mn{Fe}_{2}O_{4}$ and ${Mn}_{3}O_{4}\to Mn{Fe}_{2}O_{4}$, as shown in Figure 2.

## 5. Conclusions

Thermodynamic predictions in the form of chemical potential diagrams provide sound guidelines to assess the feasibility of the solid-state synthesis of manganese ferrite from different precursors, including oxides, metals, carbonates, hydroxides or oxy-hydroxides, and their combinations; this is useful to avoid undue experimental work with unsuitable precursor mixtures and provides suitable recommendations for subsequent heat treatments in air or controlled atmospheres for conditions where mechanical activation alone is ill suited to induce the conversion of reactants. Direct mechanosynthesis was predicted for representative combinations of oxide + oxide or oxide + metal precursor mixtures with oxygen balance, and the experimental results confirm the thermodynamic guidelines for the reactant mixture of $MnO+{Fe}_{2}O_{3}$. In this case, mechanosynthesis at room temperature yielded a slightly nonstoichiometric ferrite and residual fractions of reactants (≈16% ${Fe}_{2}O_{3}$ and ≈2% MnO) after 9 h. On the contrary, precursors without oxygen balance require a combination of mechanical activation and subsequent calcining under controlled thermochemical conditions, as confirmed experimentally for $MnO_{2}+{Fe}_{2}O_{3}$. In this case, this may still be obtained by subsequent thermal treatments in a neutral atmosphere at T ≥ 750 °C, as indicated by the thermodynamic predictions and confirmed experimentally. The formation of $Mn{Fe}_{2}O_{4}$ from the oxide + carbonate mixture was predicted by metastable diagrams in contact with $CO_{2}$-based atmospheres and was also confirmed by the experimental results. In this case, kinetic restrictions may hinder the conversion of reactants under mild mechanical activation, but the ferrite phase still forms when increasing the balls/powder ratio and using a fraction of heavier balls. Otherwise, the full conversion of reactants to $Mn{Fe}_{2}O_{4}$ may still require subsequent calcination at ≥650 °C.

## Figures and Tables

**Figure 1 materials-17-00299-f001:**
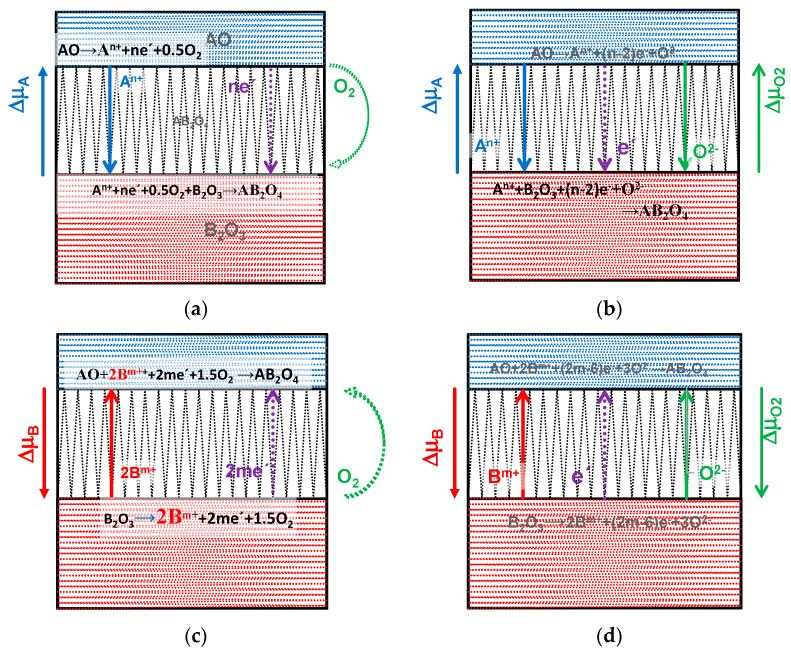
Schematic representation of mechanisms of solid-state synthesis of AB_2_O_4_ spinels from AO and $B_{2}O_{3}$ precursors under the corresponding gradients of chemical potentials, controlled by the transport of a cationic species $A^{n+}$ (**a**,**b**) or $B^{m+}$ (**c**,**d**), with ready gas phase transport in a porous reacting medium (**a**,**c**), or under additional limitations of gas phase transport and transport of oxide ions (**b**,**d**).

**Figure 2 materials-17-00299-f002:**
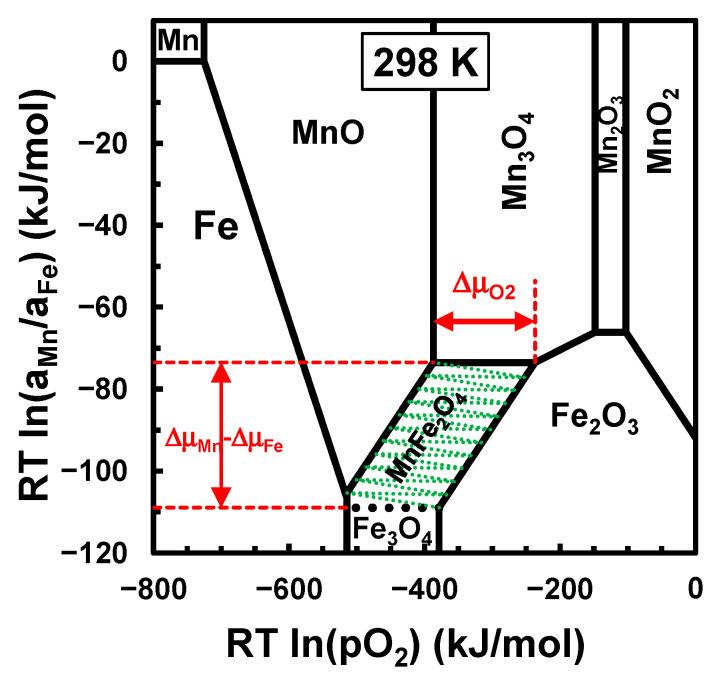
Chemical potential diagrams for the Mn−Fe−O system at room temperature.

**Figure 3 materials-17-00299-f003:**
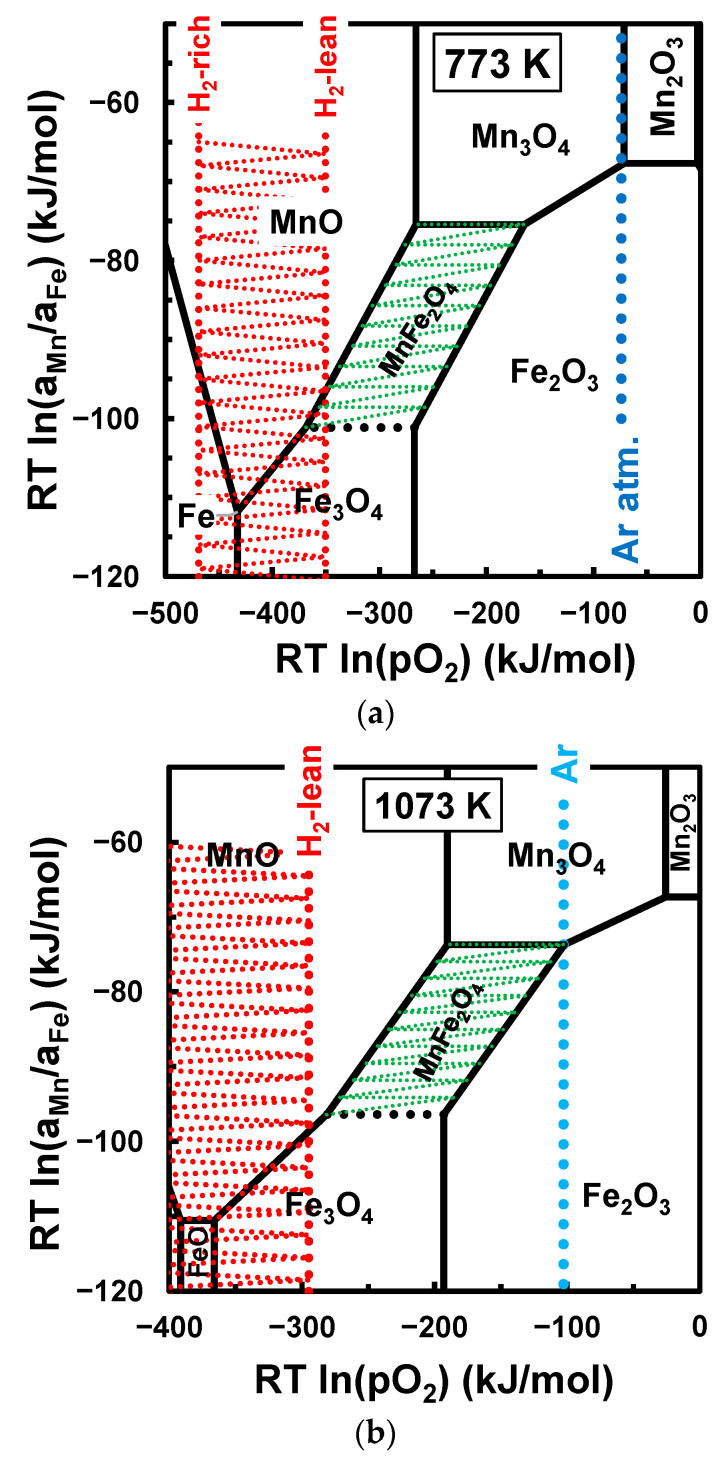
Chemical potential diagrams for the reactivity of different combinations of oxide and/or metallic precursors at 773 K (**a**), 1073 K (**b**), and 1373 K (**c**). The shaded green area shows the stability range of manganese ferrite, and the shaded red area is the redox range which may be adjusted with the $H_{2}/H_{2}O$ redox pair, from $H_{2}$-rich conditions $H_{2}:H_{2}O=100$ to $H_{2}$-lean conditions $H_{2}:H_{2}O=0.01$.

**Figure 4 materials-17-00299-f004:**
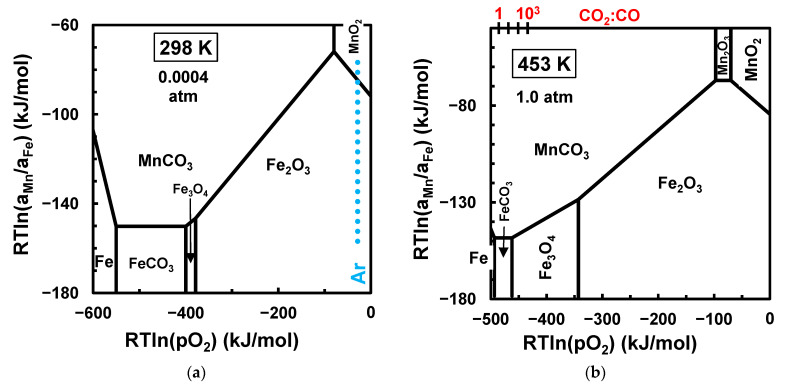
Chemical potential diagrams for the stability of iron and manganese carbonates and their equilibrium with corresponding oxides at 298 K with $p{CO}_{2}=0.0004\;atm$ (**a**) and at 453 K (**b**), 560 K (**c**), and 643 K (**d**) with $p{CO}_{2}=1.0\;atm$.

**Figure 5 materials-17-00299-f005:**
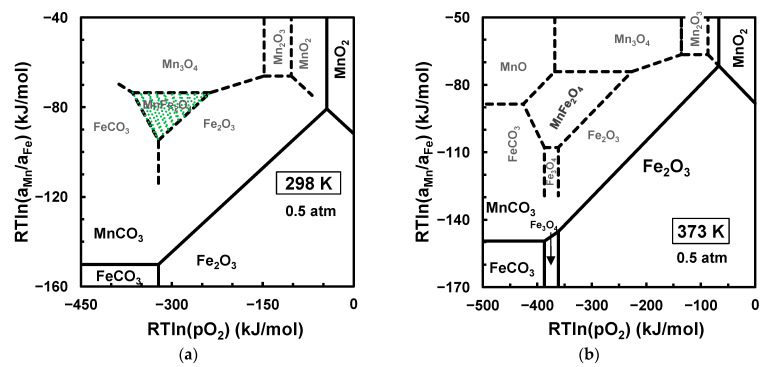
Chemical potential diagram for the Mn−Fe−O−C system at $p{CO}_{2}=0.5\;atm$ at room temperature (**a**) and at 373 K (**b**). The solid lines show equilibrium diagrams, and the dashed lines show metastable predictions for reactivity between $FeCO_{3}$ and $MnO_{2}$, assuming that the carbonation of manganese oxide is hindered.

**Figure 6 materials-17-00299-f006:**
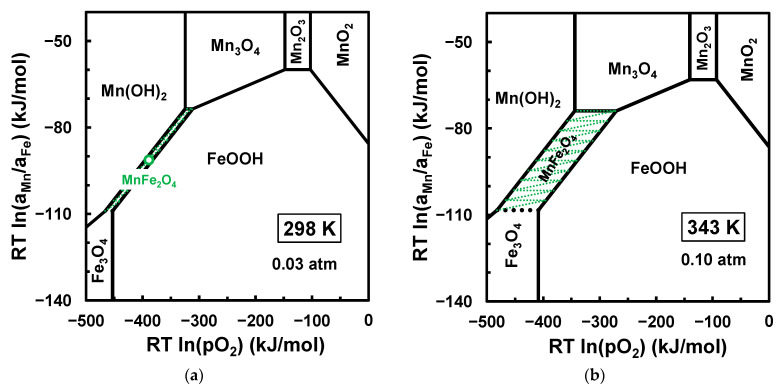
Chemical potential diagram for the Mn−Fe−O−H system at 298 K with $pH_{2}O=0.03\;atm\;(a)$ and 343 K with $pH_{2}O=0.10\;atm$ (**b**).

**Figure 7 materials-17-00299-f007:**
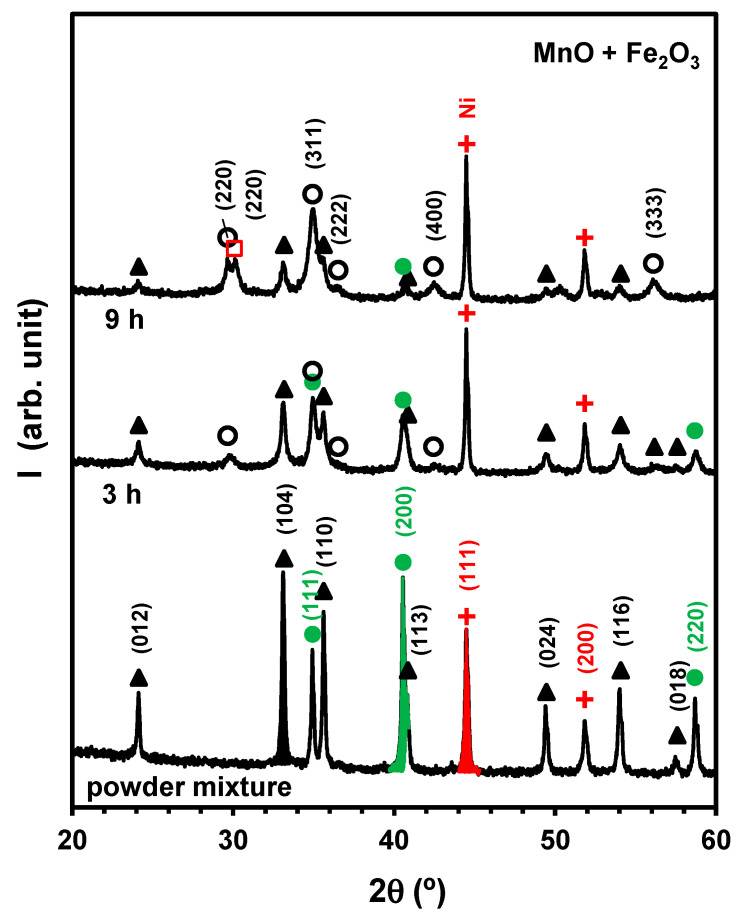
X-ray diffractograms of samples obtained by mechanosynthesis of $MnO+{Fe}_{2}O_{3}$powder mixtures milled at 500 rpm for 3 h and 9 h. Ni powder was used as an internal standard as a guideline for differences in relative intensities of residual precursor fractions and onset of reaction products. The markers identify reflections ascribed to MnO (
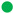
), Fe2O3 (
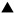
), MnFe2O4 (
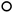
), Fe3O4 (
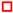
) and Ni (
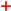
). The relative areas under the main reflection of ${Fe}_{2}O_{3}$ (104), MnO (200), and Ni (111), are shown in black, green, and red, respectively.

**Figure 8 materials-17-00299-f008:**
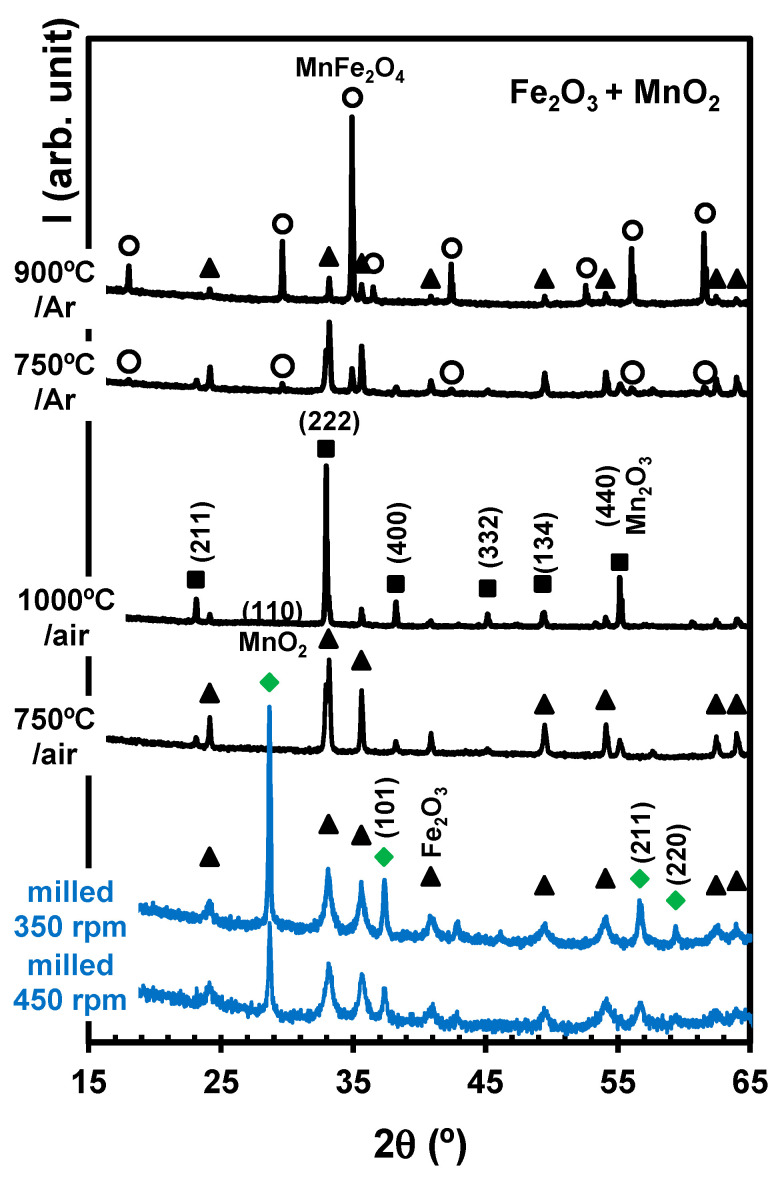
X-ray diffractograms of samples obtained by mechanical activation of ${MnO}_{2}$+${Fe}_{2}O_{3}$ powder mixtures milled at 350 rpm or 450 rpm for 10 h and then fired in air or Ar at different temperatures for 2 h. The markers identify reflections ascribed to $MnO_{2}$ (
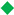
), Fe2O3 (
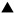
), MnFe2O4 (
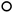
), and ${Mn}_{2}O_{3}$ (
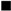
).

**Figure 9 materials-17-00299-f009:**
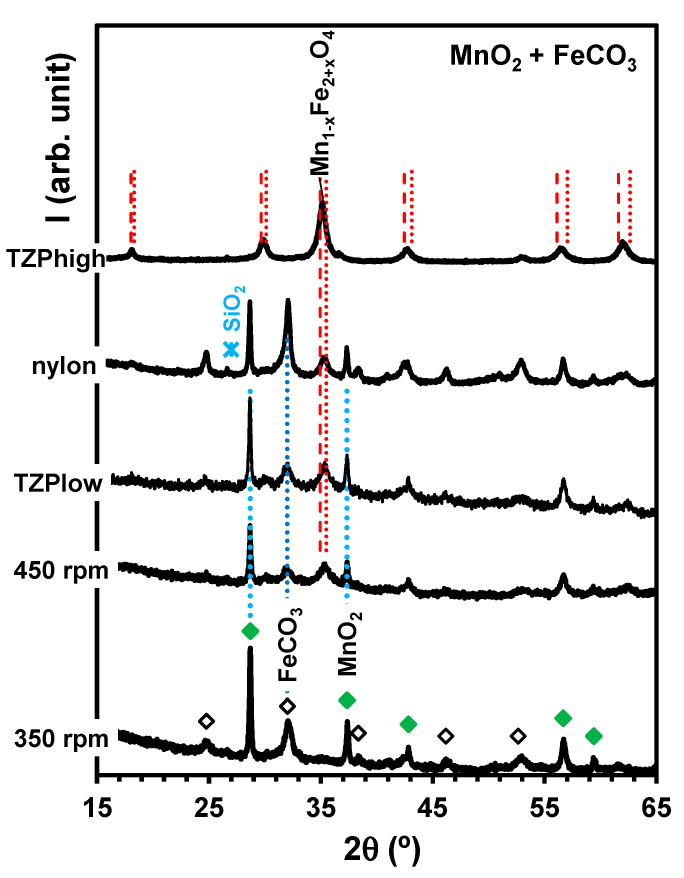
X-ray diffractograms of samples processed from $MnO_{2}+FeCO_{3}$-based siderite milled under the conditions listed in Table 1. The vertical lines show expected reflections of the most relevant phases as a guideline for conversion to the spinel phase. The markers identify reflections ascribed to $MnO_{2}$ (
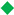
), FeCO3 (
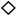
) and $SiO_{2}$ (
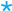
).

**Figure 10 materials-17-00299-f010:**
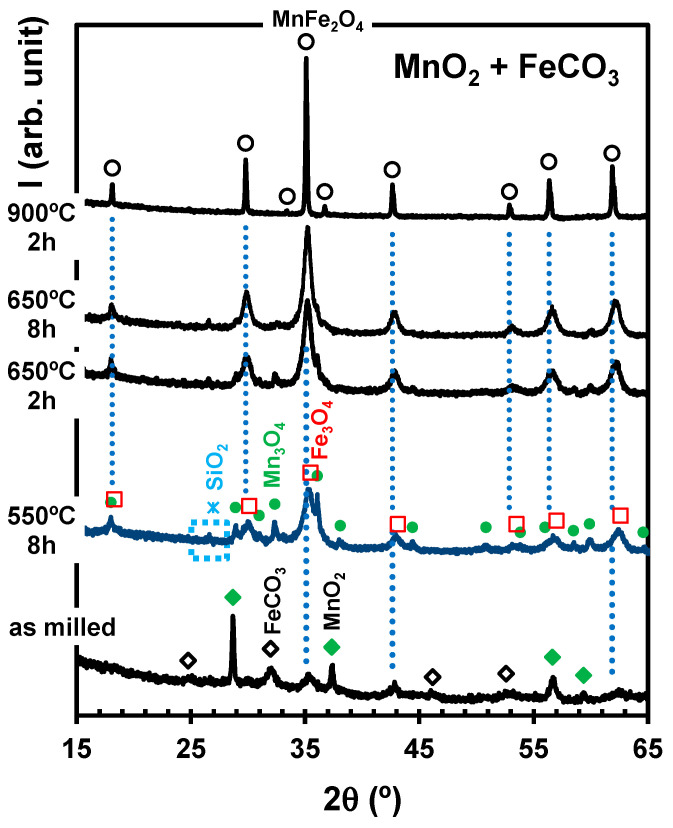
X-ray diffractograms of samples processed from $MnO_{2}+FeCO_{3}$-based siderite milled at 450 rpm and then fired at different temperatures for 2 h or 8 h in Ar atmosphere. The markers identify reflections ascribed to $MnO_{2}$ (
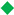
), FeCO3 (
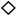
), MnFe2O4 (
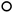
), ${Mn}_{3}O_{4}$ (

), Fe3O4 (
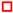
), and $SiO_{2}$ (
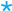
).

**Table 1 materials-17-00299-t001:** Conditions of mechanical activation or mechanosynthesis of $MnO_{2}$ + siderite powder mixtures.

Notation	Vial	% of Balls		Ball-to-Powder Weight Ratio	Rotation Speed	Effective Milling Time	Milling Time:Pause Time
	Material	10 mm	15 mm	(m_balls_:m_powder_)	(rpm)	(min)	(min:min)
350 rpm	TZP	100%	-	7:1	350	600	5:5
450 rpm	TZP	100%	-	7:1	450	600	5:5
TZP low	TZP	92%	8%	7:1	450	600	5:5
TZP high	TZP	67%	33%	12:1	450	800	10:5
Nylon	Nylon	67%	33%	12:1	450	800	10:5

**Table 2 materials-17-00299-t002:** Approximate contents of reactants and other phases (mainly the $Mn{Fe}_{2}O_{4}$-based spinel) after milling the $MnO+{Fe}_{2}O_{3}$ mixture.

Phase	*wt*.%		
	As-Milled	Milled	
		3 h	9 h
MnO	31	15	2
Fe_2_O_3_	69	37	16
Others	−	32	65

## Data Availability

Data are contained within the article.

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
