# Peer review of "Thermodynamic Guidelines for the Mechanosynthesis or Solid-State Synthesis of MnFe_2_O_4_ at Relatively Low Temperatures"

_materials, 2024, doi:10.3390/ma17020299_

Round 1
Reviewer 1 Report
Comments and Suggestions for Authors
The manuscript “Thermodynamic guidelines for mechanosynthesis or solid state synthesis of MnFeO4 at relatively low temperatures” studies the interest of a thermodynamic study to choose the best precursors for synthesis. The work is very interesting. The method is well detailed and validated by an experimental work. Minor revisions are required and are listed below.
General remarks
As lot of chemical equations are considered, the authors should indicate the free enthalpy of each one (DG).
In figures 1, 3, 4, 5 and 6 the authors should name the pictures with a, b, c, ... characters to clarify the figures and describe each picture in the legend. In the text, the authors should refer to figure XX.a,… for clarification of the manuscript.
Section 3: thermodynamic guidelines
P5, l176: Can the authors explain how they calculate the driving force 35kJ/mol?
P9, l309: in the figure 5, there is no boundary between FeCO3/Mn2O3. Do the author mean boundary FeCO3/MnO? Some clarifications are necessary in this part.
Section 4: experimental validation
P11, 349: the authors should replace “numeric percentage“ by “percentage number”.
Figure 7: Do the authors perform XRD on powders? It is not clear. Moreover, diagrams are acquired between 2q 10-90°, but only the 15-65° range is presented.
The authors use an internal standard (Ni) for XRD, however there is no quantification of the phases in the different synthesis ways. Maybe, quantification can be added to discuss these results with thermodynamic data.
P11-12: between MnO+Fe2O3 and MnO2+Fe2O3 syntheses, the authors modify the experimental procedure. Can the authors explain if it can influence the results?
P11-12: from figure 7, it seems that the spinel phase can form at room temperature after 9h or milling while it was concluded form thermodynamic calculations that the phase can not form at temperatures lower than 800°C in Ar or air. Can the authors explain this result?
Reviewer 2 Report
Comments and Suggestions for Authors
In the present work, authors reported a thermodynamic assessment to screen suitable precursors for solid state synthesis of manganese ferrite, and then investigated it structure and property. Results indicated that the thermodynamic guidelines guided the synthesis of manganese ferrite from the oxide and/or metallic precursors, powder mixtures, and other precursors. Overall, this work provides a promising method and has curtain reference function. However, some issues should be addressed.
1, The narrative in Abstract section was too tedious. The abstract should include the most important findings. Authors may focus on the what you have done and the related description should be refined to highlight your viewpoints. In addition, authors should figure out the significance and real-life application of the present paper.
2, The main objective of the paper must be written on the clearer and more concise way at the end of introduction section. Example: The objective of the present research is to study the effect of TiO2 on the surface spin disorder and inter-particle magnetic interaction of MgFe2O4 NPs which may be beneficial for different magnetic applications. Furthermore, the structural, morphological and magnetic properties of magnesium ferrite-based Titanium Oxide nanocomposite were checked and addressed.
3, Authors may rearrange/polish the text and elaborated " Experimental Method" section the way so anybody can repeat the procedures, like a recipe. If there is process flow diagram can be added in fig. 1, it would be helpful to non-specialist readers (optional).
4, During the processing, please further explain the reasons for the thermal condition selected in this paper.
5, Controllable processing is a determinant in structure and property of ferrite. How to improve the controllability and designability of ferrite particle preparation in this work? The authors should also pay attention to this challenge, and some pioneering and original researches about controllable processing of ferrite are suggested: Journal of Materials Chemistry C, 2016, 4, 9738; Nano-Micro Letters, 2023, 15, 76.
6, Page 6, line 1, it is said that “However, the mechanisms of this reaction may be complex since FeO is a very unstable phase, and may oxidise readily to 𝐹𝑒3𝑂4 in contact with the strongly oxidising phase MnO2, which may also undergo gradual reduction through intermediate phases (𝑀𝑛2𝑂3 and 𝑀𝑛3𝑂4) and down to the 𝑀𝑛3𝑂4/𝑀𝑛𝐹𝑒2𝑂4 interface in Fig.2.” Please give more details.
7, In figure 7, authors should index all these characteristic peaks. In addition, the corresponding standard JCPDS number.
8, Page 13, line 398, it is claimed that “…possibly combined with the gangue components from the natural siderite precursor.”. Please describe more and refer to literature to support it.
Reviewer 3 Report
Comments and Suggestions for Authors
The paper analyzes the mechanosynthesis of manganese ferrite at room temperature from various combinations of precursor mixtures or by mechanical activation and subsequent calcination at intermediate temperatures.
The manuscript is reasonably written and presentation of the experimental results seems to be suitable.
However, the manuscript could be improved before publication.
If possible, you should avoid using abbreviations in the title.
The practical usefulness of the obtained results should be highlighted.
The conclusions should focus on the results obtained and not just general discussions. You should use values to justify statements in the conclusions.
Comments on the Quality of English LanguageMinor editing of English language required.
